

# Legacy effects of historical grazing alter leaf stomatal characteristics in progeny plants

Jingjing Yin[1,2], Xiliang Li[2], Huiqin Guo[3], Jize Zhang[2], Lingqi Kong[2] and Weibo Ren[1]

[1] School of Ecology and Environment, Inner Mongolia University, Hohhot, China
[2] Institute of Grassland Research, Chinese Academy of Agriculture Sciences, Hohhot, China
[3] School of Life Science, Inner Mongolia Agricultural University, Hohhot, China

Corresponding author
Weibo Ren, 111979364@imu.edu.cn

## ABSTRACT

Grazing, one of the primary utilization modes of grassland, is the main cause of grassland degradation. Historical overgrazing results in dwarf phenotype and decreased photosynthesis of perennial plants. However, it remains unknown what the mechanism underlying of this legacy effect is, and the role of stomata in the resulting decreased photosynthesis also remains unclear. To address these questions, differences in stomatal density, length and width on both adaxial and abaxial epidermis were compared between overgrazing and ungrazed *Leymus chinensis* offspring by using rhizome buds cultivated in a greenhouse, and the correlation between photosynthetic capacity and stomatal behavior was also investigated. Our results showed that historical grazing significantly impacted phenotype, photosynthesis and stomatal traits of *L. chinensis*. The offspring plants taken from overgrazed parents were dwarfed compared to those taken from ungrazed parents, and the photosynthesis and stomatal conductance of plants with a grazing history decreased by 28.6% and 21.3%, respectively. In addition, stomatal density and length on adaxial and abaxial leaf surfaces were significantly increased; however, stomatal width on abaxial leaf surfaces of overgrazed *L. chinensis* was significantly decreased compared with ungrazed individuals. Moreover, the expression patterns of eight genes related to stomatal regulation were tested: seven were down-regulated (2–18 times) and one was up-regulated (three times). Genes, involved in ABC transporter and receptor-like serine/threonine protein kinase were down-regulated. These results suggest that legacy effects of historical grazing affect the stomatal conductance by decreasing the stomatal width in progeny plants, which thus results in lower photosynthesis. Furthermore, changes of stomatal traits and function were regulated by the inhibition of ABC transporter and serine/threonine protein kinase. These findings are helpful for future exploration of the possible mechanisms underlying the response of grassland plants to long-term overgrazing.

## INTRODUCTION

Grazing is one of the harshest environmental stresses and affects both the composition of the plant community and the soil properties of the natural grassland (*Krzic et al., 2006*;

*Byrnes et al., 2018*), and has led to severe degradation of arid and semi-arid grasslands globally (*Van Der Westhuizen, Snyman & Fouché, 2005*; *Liang et al., 2009*; *Schönbach et al., 2011*; *Vanderpost et al., 2011*). Previous studies have also shown that long-term overgrazing can change the composition of plant species from a perennial grass-dominated state to several degraded states dominated by sub-shrubs, shrubs, or annual plants in grasslands (*Wang, Wang & Wang, 2006*; *Sasaki et al., 2008*; *Lohmann et al., 2012*). Moreover, overgrazing was found to increase soil sand content, while decreasing both the macroporosity and water-holding capacity of soil (*Zhao et al., 2011*). Recently, it has been suggested that grazing-induced changes in plant traits may persist even after the grazing pressure is removed. This may therefore affect the performance of the following progeny plants, a phenomenon that is called the legacy effect (*Veen et al., 2014*; *Casas et al., 2016*).

The legacy effect has been described in ecology since the early 1990s and has been shown to be involved in plant succession, plant invasion, human land-use impacts, and herbivory or grazing impacts (*Simard, 1995*; *Cuddington, 2011*). The legacy effect of herbivory, which means that herbivory-induced effects in the parental generation can result in the modification of offspring plants, has recently received increasing attention (*Agrawal, Laforsch & Tollrian, 1999*; *Poma et al., 2014*). Herbivory-induced effects may persist across one or more generations of the offspring via sexual or asexual reproduction (*Yin et al., 2019*). Generally, these legacy effects are often considered adaptive, particularly when they trigger pre-adaption in offspring plants to similar stresses that parent plants have experienced (*Herman & Sultan, 2011*; *Holeski, Jander & Agrawal, 2012*). A meta-analysis summarizing 139 experimental studies in plants and animals, focusing on 1,170 effect sizes, showed that transgenerational effects benefit offspring plants in response to stressful conditions (*Yin et al., 2019*). Moreover, legacy effects can be enabled by altering seed quality (*Herman & Sultan, 2011*) or by epigenetic modification (*Holeski, Jander & Agrawal, 2012*). For example, drought stress has been shown to have negative effects on seed quality by reducing germination and seedling vigor in the F1 generation (*Wijewardana et al., 2019*). *Boyko et al. (2010)* showed that stress-induced trans-generational effects were regulated by changes in DNA methylation and small-RNA silencing in *Arabidopsis thaliana*.

Over the past few decades, much research has been conducted to analyze the ecological and physiological aspects of grazing stresses on natural grasslands (*Zhang et al., 2017*; *Dong et al., 2018*; *Mueller et al., 2017*). Recently, studies on the mechanism of degeneration from individual traits have increased. It has been shown that long-term overgrazing led to the miniaturization of plant individuals, causing for example, stunted height, shortened and narrowed leaves and similar effects (*Wang et al., 2000*). At the same time, physiological traits such as photosynthetic rate also decreased severely after long-term overgrazing, which were still maintained in offspring plants (*Ren et al., 2017*). However, little is known about how this legacy effect on phenotypic characteristics is maintained. Therefore, this article explored likely underlying mechanisms based on the stomatal behavior and function.

Stomata, the small pores on the surfaces of leaves and stems, formed by a pair of guard cells, play a critical role in the balance of $CO_2$ uptake for photosynthesis and water loss

through transpiration. It is interesting to note that the area of the total stomatal pore may only be 5% of a plant leaf surface (*Fricker & Willmer, 1996*). Moreover, previous studies have suggested that photosynthesis is positively correlated with stomatal conductance (gs) (*Wong, Cowan & Farquhar, 1979*; *Lawson et al., 2018*), and maximum level of gs is dictated by the density and size of stomata (*Hetherington & Woodward, 2003*; *Franks & Farquhar, 2007*). Stomatal characteristics are affected by many factors, such as $CO_2$ (*Woodward & Kelly, 1995*; *Sekiya & Yano, 2008*), water (*Yordanova, Uzunova & Popova, 2005*; *Arve et al., 2013*; *Saradadevi et al., 2017*), light (*Shimazaki et al., 2007*; *Mott, 2009*), and temperature (*Liu et al., 2018*; *Hill et al., 2014*). The correct coordination between photosynthesis and stomatal behavior is essential for different plant function. Numerous studies have explored the effects of herbivory on photosynthesis (*Chen et al., 2005*; *Peng et al., 2007*; *Kerchev et al., 2012*). In contrast, little is known about the role of stomata in this process. The genetic regulation of stomatal development and movement was mainly studied in the model plant *A. thaliana* (*Nadeau & Sack, 2002*; *Vráblová et al., 2017*); however few studies attempted to study this in forage plants. Therefore, this topic is worthy of further study and it is particularly interesting which genes participate in the regulation of the stomata response to overgrazing.

*Leymus chinensis* is a perennial, rhizomatous clonal grass with high palatability and forage value (*Liu, 2006*), that largely covers the eastern Eurasian temperate grassland area. As the dominant grass species of the grasslands, *L. chinensis* plays an important role in the grassland ecosystem. The present study focused on *L. chinensis* and investigated the effects of historical overgrazing on phenotype, photosynthesis and leaf stomatal characteristics of offspring plants. Previous studies showed that a history of overgrazing can lead to dwarfing in plants and significantly decreased photosynthesis (*Ren et al., 2017*). However, little is known about the regulating mechanisms of this decrease in photosynthesis. As an important organ of photosynthesis, it is unknown whether stomata play a key role in this process. Therefore, two questions were addressed by this research: (i) Do legacy effects induced by historical grazing change the characteristics of leaf stomata in the following generation? (ii) Which genes take part in the regulation of the stomata response to overgrazing?

## MATERIALS AND METHODS

### Field site and plant material

The experimental materials were taken from an artificial grassland located at the field station of the Grassland Research Institute of the Chinese Academy of Agricultural Sciences, located in Hohhot, Inner Mongolia, China (N: 40°36′; E: 111°45′; H: 1,065 m). The artificial grassland was established for the field research in July 2009, and planted with the forage species *L. chinensis*, *Elymus nutans Griseb.*, and *Medicago sativa L. cv. Aohan.*, *Emanmus sibiricus L.*, etc. The two treatments included overgrazing (GZ) and no grazing (NG) in this grazing experiment plot, three replications were conducted for each treatment, and the area of each plot was about 0.64 hm². For grazing treatments, six sheep continuously grazed in the GZ plot during the growing season from May to October each year at a stocking rate of about nine sheep per hectare. This study selected the rhizome

buds of *L. chinensis* as offspring generation to examine the legacy effects of historical grazing in parental plants.

## Greenhouse experiment

Fresh grass had not germinated at the early spring of 2018, and the rhizome buds of *L. chinensis* were randomly taken from both GZ and NG plots in the artificial grassland. The rhizomes coming from different parental plants were cut to 2–3 cm lengths and each rhizome with a bud was planted into a pot and transferred to the greenhouse for cultivation. The buds were kept in the greenhouse for 7 weeks. Soil was taken from the grazing experiment plots, and was sieved and mixed to ensure uniformity of the soil. The rhizomes were planted into pots with a diameter of 18 cm after the pots were filled with soil. Each treatment used seven pots with three rhizomes per pot. In total, 14 pots were randomly arranged in the greenhouse. The temperature was about 25 °C during the day and 15 °C at night. The phenotypic traits, gas exchange parameters, stomatal traits, and related gene expression studies were successively tested after plant transplantation based on the following methods.

## Morphological traits

The plant height was measured from the 2nd week after transplantation and was continually tested every week. The leaf length (LL) and leaf width (LW) of the second leaf from the top of each individual were measured at week seven of growth in the greenhouse. The leaf area was calculated according to the formula: Area = 0.655* (LL × LW) (*Yajun, 2009*).

## Gas exchange measurements

After about 7 weeks of culture, the gas exchange parameters of the second leaf from the top of each plant were measured using a LI-6400 (LI-COR, Lincoln, NE, USA) photosynthetic measurement system during a clear morning from 9:00 to 11:00. Measurement indicators included: net photosynthetic rate ($P_N$), transpiration rate ($E$), inter-cellular $CO_2$ concentration ($Ci$), stomatal conductance ($gs$), and water use efficiency ($WUE$) calculated from photosynthetic rate and transpiration rate ($WUE = P_N/E$). The gas exchanges were measured as photosynthetically active radiation ($PAR$) of $1000 \pm 10$ $\mu$mol m$^{-2}$ s$^{-1}$ and $CO_2$ concentration of $350 \pm 3$ ppm.

## Stomatal traits

After about 7 weeks of culturing, the second leaf of each of the treated plants of *L. chinensis* was collected to determine the stomatal traits using the impression approach (*Xu & Zhou, 2008*). After collection, the leaves were rapidly fixed with 2.5% glutaraldehyde, and brought back to the laboratory. The abaxial and adaxial epidermis of the leaf were first cleaned using a paper towel, and then carefully smeared with transparent nail polish in the tail, middle, and top area, gently peeled off after approximately 15 min, then immediately mounted on a glass slide, covered with a cover slip, and finally observed and photographed using an optical microscope (BX53, Olympus, Tokyo, Japan). To count the number of stomata in adaxial and abaxial leaf surfaces in the microscope at

40× magnification, and to calculate the stomatal density (stomatal density = number of stomata/area in view), five fields of view were randomly observed for each piece. Seven stomata were randomly selected from the microscopic slides, and stomatal length and width in the microphotographs were analyzed with Image J 1.0 image processing software (National Institutes of Health, Bethesda, MD, USA). Since the degree of opening of the pores changes constantly, the length and width of guard cells were selected to represent the maximum degree of opening of the pores, that is, the size of the pores (Fig. S1).

### Real-time PCR analysis

Samples were taken from the second or third leaf of offspring plants treated with enclosure and overgrazing of parental plants after about 7 weeks of culture, then stored at −80 °C. RNA isolation used the TRIZOL reagent (Invitrogen, Carlsbad, CA, USA) following the procedures provided by the manufacturer. Then, the quantity of RNA was evaluated by electrophoresis on a 1.0% agarose gel, and the concentration was measured by an ultraviolet spectrophotometer (UV-2550PC, Shimadzu, Japan). High-quality total RNA samples were used for subsequent experiments. According to the manufacturer's instructions, any possible DNA contamination was eliminated from RNA samples, and first-stand cDNA was synthesized by using the Prime Script $^{RT}$ reagent Kit with gDNA Eraser (Takara, Tokyo, Japan). The cDNA samples were stored at −20 °C for real-time PCR analysis. In addition, eight differentially expressed genes related to stomatal development and regulation were selected based on transcriptome data (Ren et al., 2018). The program primer premier 5 was used to design the required primer sequences, and the primer details of these genes are presented in Table S1 of the Supporting Information. The real-time PCR for gene expression was performed on Quant Studio$^{TM}$ 6 Flex System Software using SYBR Green 1 reagents, following the instructions of test kits.

### Statistical analyses

All statistical analyses were conducted in Microsoft Excel, and independent samples $T$ tests were performed to assess the differences between treatments for phenotypic characteristics, photosynthetic parameters, and stomatal traits at the $P < 0.01$ and $P < 0.05$ levels of significance using SPSS 19.0 statistical software (Systat Software Inc., Chicago, IL, USA). Graphical representations were generated with Sigmaplot 13.0 software (Systat Software Inc., San Jose, CA, USA).

## RESULTS

Historical grazing on parent plants significantly affected the growth of offspring plants (Figs. 1 and 2). Offspring developed from grazed parents grew much slower than those from ungrazed parents (Fig. 1B). Compared with the ungrazed control, the plant height and leaf length of grazed progeny decreased by 23.5% and 15.4% (Figs. 1 and 2A), the leaf width and area of grazed progeny decreased by 9.7% and 23.1% respectively (Figs. 2B and 2C), and the aboveground biomass of individual grazed plants also decreased by 12.9% (Fig. 2D).

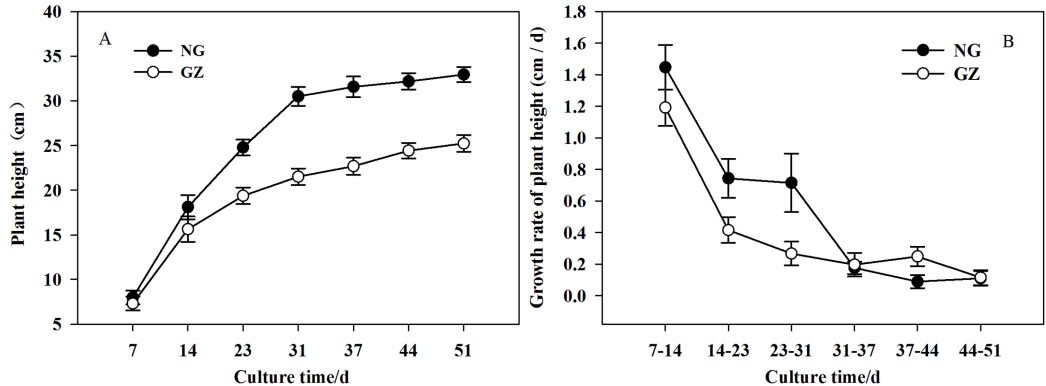

**Figure 1 The differences in growth of offspring plants developed from overgrazed and ungrazed** *Leymus chinensis* **respectively in the culture period (Mean ± SE).** (A) The change of plant height with culture time; (B) The change of growth rate of plant height with culture time. NG, Ungrazed; GZ, Grazed.

Historical grazing has also significant impact on photosynthesis. $P_N$ and $gs$ in progeny plants generated from grazed parents were significantly decreased by 28.6% and 21.3%, respectively, compared with control plants ($P < 0.05$, Figs. 3A and 3B). However, $Ci$ in ungrazed progeny plants increased and $WUE$ decreased significantly ($P < 0.05$, Figs. 3C and 3E).

Stomatal density on the basal adaxial leaf surfaces in progeny plants of grazing-stressed *L. chinensis* increased significantly in the greenhouse experiment (Fig. 4A). However, no significant differences were found both in the middle and top leaf surfaces (Figs. 4B and 4C). Noticeably, historical grazing significantly increased stomatal density and length; however, the stomatal width on the basal of abaxial leaf surfaces decreased (Figs. 4D–4F). Moreover, stomatal density and width on the middle of abaxial leaf surfaces in offspring plants taken from grazed parents decreased significantly compared with those taken from ungrazed parents (Figs. 4D–4F). Notably, no significant differences were found in stomatal characteristics on the top of adaxial and abaxial leaf surfaces (Figs. 4D–4F), suggesting that stomatal differences were mainly found at the basal and middle of leaves, not at the top of leaves.

To explore the possible mechanism of stomatal differences induced by historical grazing, the expressions of eight key genes related to stomatal regulation were investigated (Table 1). These genes are classified into two types based on the pathways they are involved in: (1) stomatal movements (*ABCCL4*, *ABCG5*, *SAPK10*, and *GNAT1*); (2) stomatal development (*ER1*, *ER*, *ERL1*, and *CYCP3L-1*) (Table 1). The results showed that all genes of grazed progeny plants were down-regulated by 2–18 times except for *CYCP3L-1* which was up-regulated (Figs. 5A–5H). This suggests that stomatal characteristics were mainly affected by the down-regulation of related genes.

## DISCUSSION

### The effects of historical grazing on stomatal characteristics

Plants can alter stomatal activity to cope with environmental stresses such as drought (*Changhai et al., 2010*), cold (*Reynolds-Henne et al., 2010*), insects (*Papazian et al., 2016*),

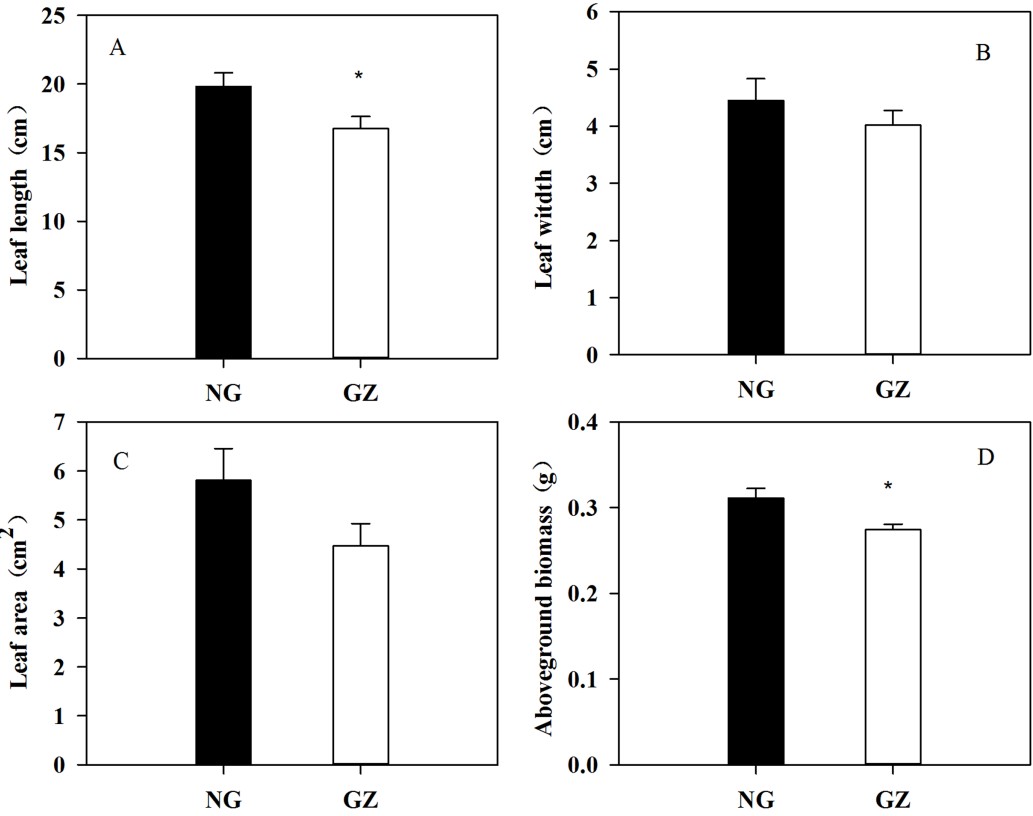

**Figure 2 Effects of historical grazing on leaf length (A), width (B), area (C), and aboveground biomass (D) in offspring of *Leymus chinensis* (Mean ± SE).** NG, Ungrazed; GZ, Grazed. The asterisk indicates a significant difference between the grazing and ungrazing treatments. *$p < 0.05$.

and even grazing (*Zhang, 2010*). This study provides clear evidence that the grazing history can impress legacy effects on traits and functions of leaf stomata in *L.chinensis*. After grazing, stomatal density increased significantly on the basal of adaxial and abaxial leaf surfaces and decreased significantly on the middle of abaxial leaf surfaces. Similar findings have been reported for *Thalictrum aplinum*, *Kobresia humilis*, *Gentiana straminea*, and *Elymus nutans Griseb* (*Zhang, 2010*). Moreover, stomatal length of the progeny developed from grazed parents increased, while stomatal width decreased significantly in abaxial leaf surfaces. These data suggested that stomata of progeny plants tended to close because of the long-term overgrazing experience of their parental plants, which resulted in a decrease in gs. A similar result of stomatal closure caused by herbivory was also reported previously (*Papazian et al., 2016*). Hence, it may be an alternative strategy to close the stomata to adapt to grazing stress.

Stomata are deliberately regulated by a series of genes to help plants adapt to environmental stimuli. This study investigated eight key genes that play key roles in the development and movement of stomata. It has been suggested that historical grazing affects the expression level of these genes on the progeny developed from grazed parents. After long-term grazing, the expression of seven of these eight tested genes, mainly related to stomatal development and regulation, were significantly decreased. For example,

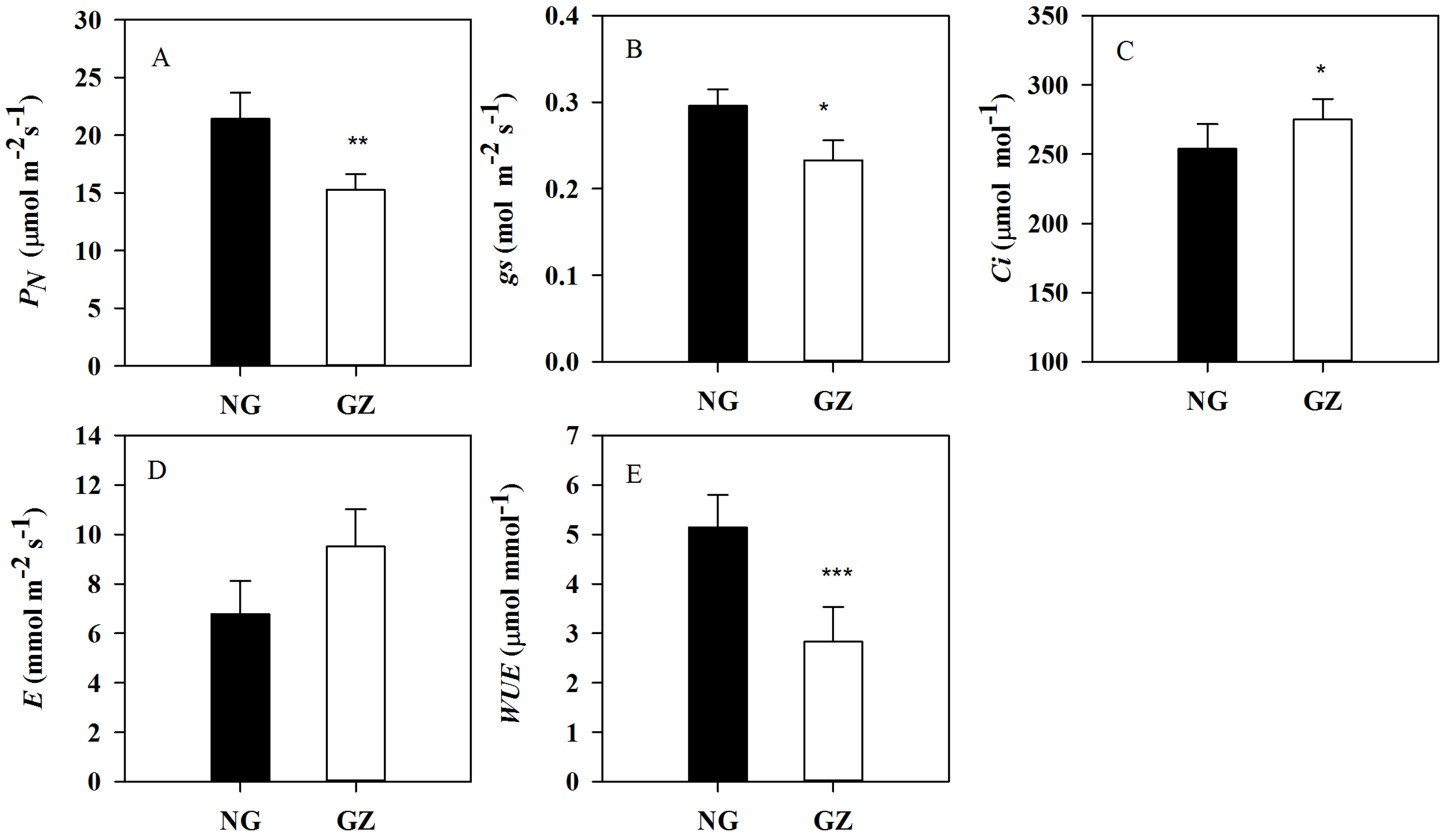

**Figure 3 Effects of historical grazing on leaf photosynthetic characteristics in progeny plants (Mean ± SE).** $P_N$, net photosynthetic rate; $gs$, stomatal conductance; $Ci$, intercellular carbon dioxide concentration; $E$, transpiration rate; $WUE$, water use efficiency. NG, Ungrazed; GZ, Grazed. Different asterisks indicate a significant difference between the grazing and ungrazing treatments. $^*p < 0.05$; $^{**}p < 0.01$; $^{***}p < 0.001$.

*ABCCL4* and *ABCG5* are two members in the superfamily of the ATP-binding cassette (ABC) transporters, which has more than 120 members in both rice (*Oryza sativa*) and *A. thaliana*. Stomatal regulation is one of the fundamental processes in which ABC transporters participate (*Rea, 2007*). A previous study supported that ABC transporter *AtMRP5* mutants tended to have more closed stomata than wild-type control plants, resulting in decreased transpiration rate and increased drought tolerance (*Klein et al., 2003*). Therefore, the down-regulation of the expressions of *ABCCL4* and *ABCG5* may also be relevant to closing stomata. In addition, *GNAT1* plays an important role in signal transmission, as G proteins participate in ethylene-induced stomatal closure and functions through hydrogen peroxide synthesis in *A. thaliana* (*Ge et al., 2015*). Moreover, in this study, three cell-surface-resident receptor-like kinases genes (*ER1*, *ER*, and *ERL1*) were detected, which encode members of the large family of plant LRR receptor-like kinases (RLKs) (*Shpak et al., 2004*), which play a critical role in the regulation of stomatal density. *Shpak (2005)* identified the ER-family LRR-RLKs as negative regulators of stomatal development, and the down-regulation of *ER* genes and the up-regulation of *CYCP3L-1* provide molecular evidence for increased stomatal density and length on the basal of adaxial and abaxial leaf surfaces. Another serine/threonine–protein kinase gene (*SAPK10*),

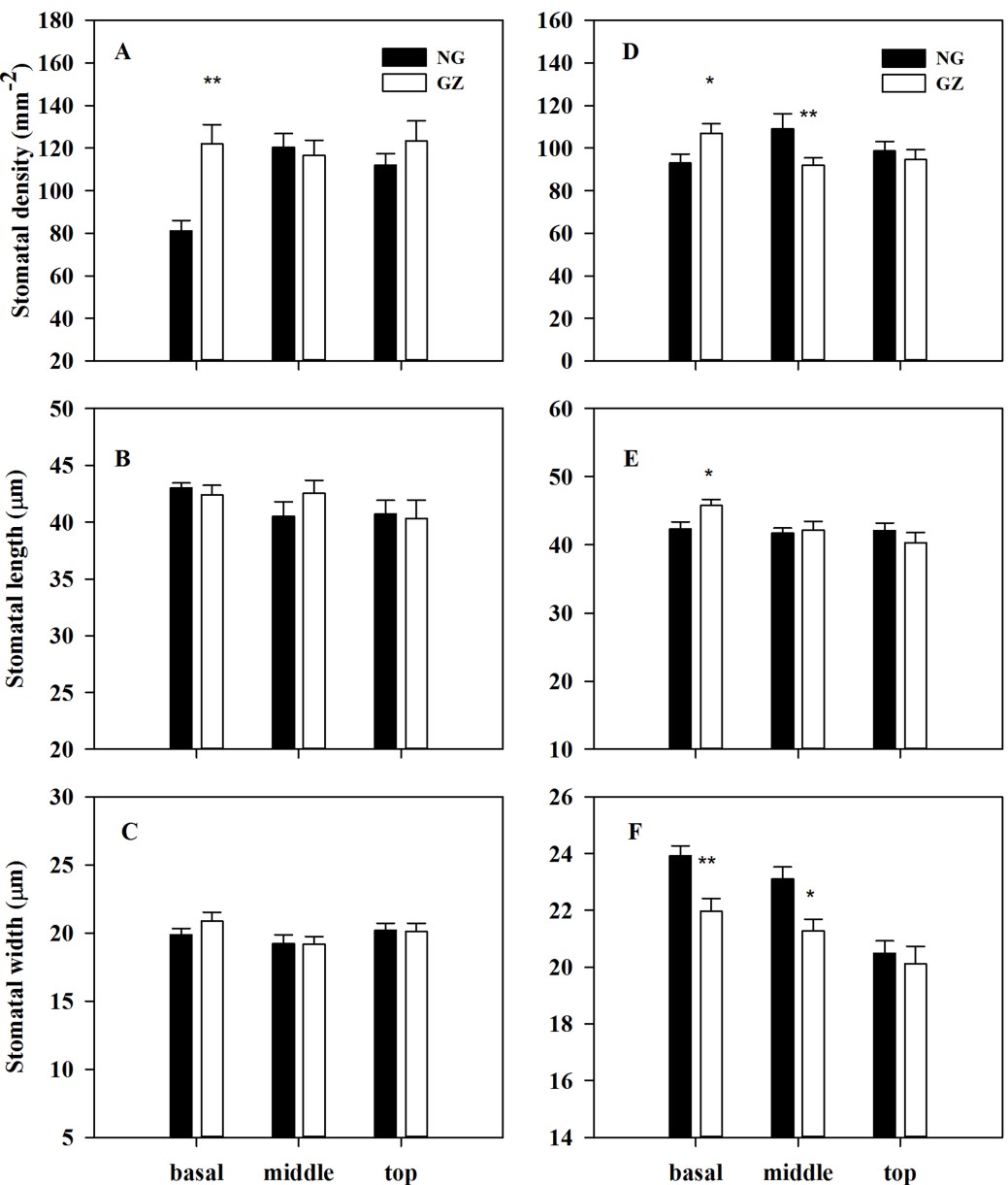

**Figure 4** **Effects of historical grazing on stomatal characteristics (stomatal density, length and width) of adaxial (A–C) and abaxial (D–F) leaf surfaces in progeny plants of *Leymus chinensis* (Mean ± SE).** NG, Ungrazed; GZ, Grazed. Different asterisks indicate a significant difference between the grazing and ungrazing treatments. $^*p < 0.05$; $^{**}p < 0.01$. 

which is specifically expressed in guard cells, plays an important role in stomatal movement (*Min et al., 2019*). However, the regulation mechanism of *SAPK10* remains unclear and requires further study.

## Legacy effects on stomata

Stomatal numbers and their pattern on the leaf surface can be either directly or indirectly influenced by the growth environment. Stomatal density is one of the key factors that may regulate the stomatal conductance and functions (*Doheny-Adams et al., 2012*). Increased

**Table 1 The expression of genes related stomatal development and regulation.**

| Gene No. | Annotation | Abbreviation | Function |
|---|---|---|---|
| >c118857.graph_c0 | ABC transporter C family member 4-like (LOC109773936) | *ABCCL4* | Stomatal movements |
| >c92989.graph_c0 | ABC transporter G family member 5 (LOC109768344) | *ABCG5* | Stomatal movements |
| >c89156.graph_c0 | Serine/threonine-protein kinase SAPK10 (LOC109758947) | *SAPK10* | Stomatal movements |
| >c103838.graph_c0 | Guanine nucleotide-binding protein alpha-1 subunit (LOC112875249) | *GNAT1* | Stomatal movements |
| >c128305.graph_c0 | Receptor-like serine/threonine protein kinase 1 (ER1) gene (JQ599260) | *ER1* | Stomatal development |
| >c93997.graph_c0 | LRR receptor-like serine/threonine-protein kinase ERECTA (LOC109740979) | *ER* | Stomatal development |
| >c113269.graph_c0 | LRR receptor-like serine/threonine-protein kinase ERL1 (LOC109783729) | *ERL1* | Stomatal development |
| >c134962.graph_c0 | Cyclin-P3-1-like (LOC109778329) | *CYCP3L-1* | Stomatal development |

**Note:**
Up, up-regulation; Down, down-regulation.

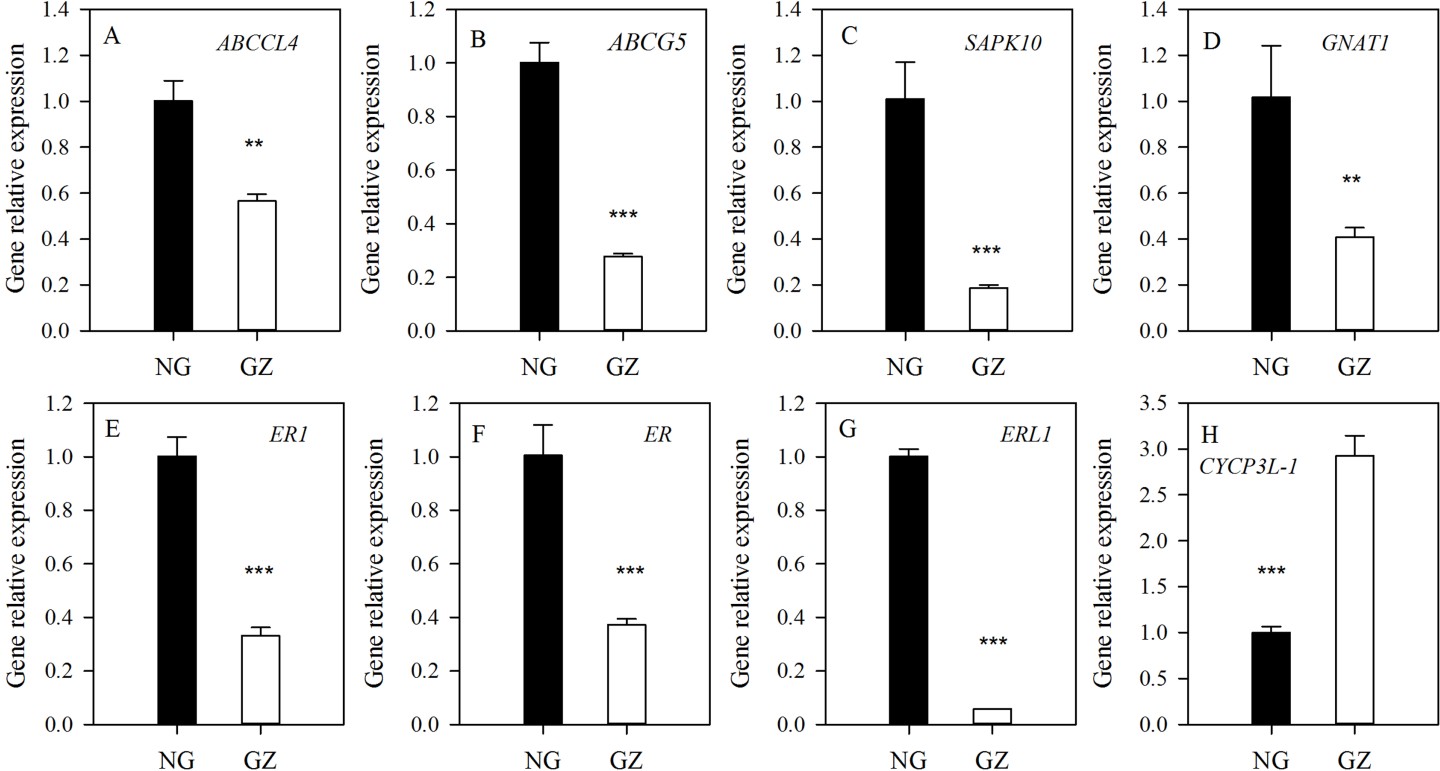

**Figure 5 Differences in expression of genes related to stomatal regulation of progeny plants developed from overgrazed and ungrazed *Leymus chinensis* respectively.** NG, Ungrazed; GZ, Grazed. Different asterisks indicate a significant difference between the grazing and ungrazing treatments. $^{**}p < 0.01$; $^{***}p < 0.001$.

stomatal density offers more potential for $CO_2$ diffusion into the leaf, which facilitates higher rates of gas exchange in responses to environmental stimuli (*Yoo et al., 2011*; *De Boer et al., 2016*). Furthermore, a close positive correlation exists between stomatal conductance and photosynthetic rate (*Buckley & Mott, 2013*; *Farquhar & Sharkey, 1982*). *Tanaka et al. (2013)* investigated *A. thaliana* mutants with altered stomatal density and showed that the photosynthetic rate could be enhanced by 30% via increasing stomatal density in *STOMAGEN*-overexpressing *A. thaliana* compared with wild-type plants. In the

present research, the photosynthetic rate decreased significantly in response to long-term grazing treatment in the parental generation of *L. chinensis*. A similar finding was also reported in previous research (*Ren et al., 2017*). The performance is the same in the subsequent generation that developed from grazed parents. However, the photosynthetic rate of offspring plants decreased by 28.6% compared with that of the un-grazed control. The possible reason is that stomatal aperture weights much than stomatal density on which gs is determined (*Weyers & Lawson, 1997*). Furthermore, changes in stomatal density can be compensated for by changes in stomatal opening or closure. For example, *Büssis et al. (2006)* used transgenic *A. thaliana* plants that over-expressed the *SDD1* gene and *sdd1-1* mutants showed that increased stomatal density could be compensated for by decreased stomatal aperture.

Many studies have showed that livestock grazing may change soil quality or properties and leads to soil compaction relative to ungrazed soil (*Daniel et al., 2002*; *Drewry, Cameron & Buchan, 2008*; *Kotzé et al., 2013*; *Pulido et al., 2016*; *Byrnes et al., 2018*). This in turn can result in decreased soil pore space, reduced infiltration, and even decrease the supply of available water (*Willatt & Pullar, 1984*; *Kotzé et al., 2013*; *Tate et al., 2004*). To address the water limitation, plants may decrease their stomatal aperture to minimize water loss. However, the closure of stomata may also decrease the uptake of $CO_2$, which may decrease the photosynthetic rate and inhibit plant growth. Hence, plants may face the trade-off between water loss and $CO_2$ uptake. For example, *Case & Barrett (2001)* investigated two populations of *Wurmbea dioica* separately in wet and dry conditions, their results showed that the stomata of the dry populations were partly closed to minimize water loss and decrease photosynthesis rates. The detailed mechanisms of the stomatal response to grazing are not easy to explain, because grazing involves a complex set of factors. However, the results of the present study suggested the existence of legacy effects of livestock grazing in the parents with regard to the stomatal behavior in offspring generation of *L. chinensis*. Moreover, the changes in stomatal characteristics decreased stomatal conductance and photosynthesis rate, which resulted in a dwarf phenotype of offspring plants (Fig. 6). So far, exceedingly little is known about the detailed mechanisms of these legacy effects; however, increasing evidence suggests that epigenetic mechanisms such as DNA methylation, histone modifications, and small RNAs contribute to these legacy effects (*Agarwal, 2001*; *Champagne, 2008*; *Richards, 2011*; *Boyko et al., 2010*). Further research is needed to fully understand the epigenetic mechanisms of how the legacy effects are induced by grazing from the parents to progeny plants.

## Legacy effects of historical grazing on progeny plants

Long-term overgrazing, which can affect the stability of the plant community by altering the growth of individual plants (*Niu et al., 2015*; *Zuo et al., 2018*), is an important cause for grassland degradation (*Bai et al., 2012*). Previous research has shown that transgenerational morphological plasticity induced by overgrazing heavily involves photosynthetic function in *L. chinensis* (*Ren et al., 2017*). In general, phenotypic plasticity refers to the ability of organisms to modify their morphology, physiology, or behavior and adapt to changing environmental conditions (*Wadgymar & Austen, 2019*). This

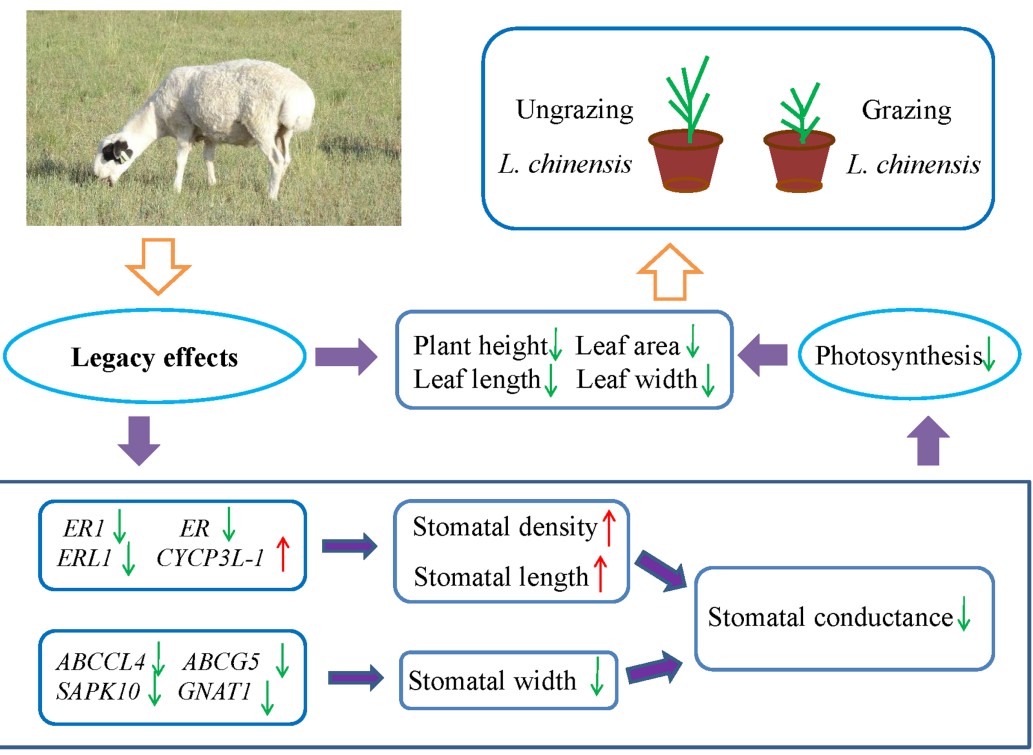

**Figure 6 Schematic diagram illustrates the regulation pathway of historical overgrazing-induced dwarf phenotype in *Leymus chinensis*.**

experiment further observed that overgrazing experienced by the parental generation could alter stomatal characteristics of the offspring of *L. chinensis*. The decreased photosynthesis and the dwarfed phenotype may be attributed to transgenerational stomatal plasticity. However, little is known about transgenerational effects induced by overgrazing in forage grass. According to a recent study by *Kafle, Wurst & Dam (2019)*, single herbivory in the parental generation did not prime the transgenerational response, only sequential above- and below- ground herbivory did. Therefore, it can be speculated that transgenerational responses require a certain threshold of environmental stress. Moreover, the developmental stage at which the parental generation is subjected to stress can also influence the performance of their offspring (*Burton & Metcalfe, 2014*). However, for perennial asexual plants, the threshold of cross-generational effects and how long it can last in the offspring generation remain unknown and require further exploration.

Transgenerational legacy effects have received increasing attention in recent years. Although only the legacy effects raised from historical grazing in *L. chinensis* was investigated in this study, the results still provide hints what may happen in other plants. *Holeski, Jander & Agrawal (2012)* have also summarized several examples of transgenerational effects of the resistance to herbivores and pathogens in 11 plant species. Previous studies have shown that the offspring generation could perform better in the maternal than in the non-maternal environment (*Galloway & Etterson, 2007*; *Latzel et al., 2014*). *L. chinensis*, as an asexual plant, consists of many interconnected ramets, where each ramet may experience different environmental conditions. This leads to better

adaptation of offspring ramets to the living environment by inheriting information from the maternal generation. Therefore, the legacy effects of historical grazing are more conducive to both the survival and growth of progeny plants.

## CONCLUSION

This study found that legacy effects of historical grazing can alter leaf stomatal characteristics in progeny plants. The results suggested the existence of a trade-off between density and aperture of stomata in *L. chinensis* in response to long-term overgrazing stress. Changes in stomatal characteristics caused decreases in gs and photosynthesis rate, which resulted in a dwarf phenotype of offspring plants. Moreover, five important genes (*ABCCL*, *ABCG5*, *ER1*, *ER* and *ERL1*), which are involved in ABC transporter and receptor-like serine/threonine protein kinase, played a key role in the process of stomatal regulation. This regulation mechanism of the process has important significance for a better understanding of the dwarf phenotype of perennial grassland plants.

## ACKNOWLEDGEMENTS

We gratefully thank Fenghui Guo for his help with fieldwork, and we also thank Xiumin Yu for the technical help during the experiment.

### Funding

This work was supported by the National Natural Science Foundation of China (No. 31872407), Major Science and Technology Project of Inner Mongolia (No. 2019ZD008) and Foundation for High-level Talents of Inner Mongolia University (No. 12000-15031908). The funders had no role in study design, data collection and analysis, decision to publish, or preparation of the manuscript.

### Grant Disclosures

The following grant information was disclosed by the authors:
National Natural Science Foundation of China: 31872407.
Major Science and Technology Project of Inner Mongolia: 2019ZD008.
Foundation for High-level Talents of Inner Mongolia University: 12000-15031908.

### Competing Interests

The authors declare that they have no competing interests.

### Author Contributions

- Jingjing Yin conceived and designed the experiments, performed the experiments, analyzed the data, prepared figures and/or tables, authored or reviewed drafts of the paper, and approved the final draft.
- Xiliang Li conceived and designed the experiments, analyzed the data, prepared figures and/or tables, authored or reviewed drafts of the paper, and approved the final draft.

- Huiqin Guo performed the experiments, authored or reviewed drafts of the paper, and approved the final draft.
- Jize Zhang performed the experiments, authored or reviewed drafts of the paper, and approved the final draft.
- Lingqi Kong performed the experiments, authored or reviewed drafts of the paper, and approved the final draft.
- Weibo Ren conceived and designed the experiments, analyzed the data, prepared figures and/or tables, authored or reviewed drafts of the paper, and approved the final draft.

## Data Availability

Raw data is available as a Supplemental File.

## Supplemental Information

Supplemental information for this article can be found online at http://dx.doi.org/10.7717/peerj.9266#supplemental-information.

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
