# Peer review of "Legacy effects of historical grazing alter leaf stomatal characteristics in progeny plants"

_PeerJ, doi:10.7717/peerj.9266_

## Round 0.1 · original submission · Major Revisions

The reviewers are in general positive to your work. Pay special attention to comment of reviewer 3 concerning the possibility to add other plant features. Please, provide a point-by- point response letter.

Reviewer 1 ·

Basic reporting

Poor English.

Experimental design

Poor design.

Validity of the findings

Not sure

Additional comments

My comments are as follows:
1. For the legacy effect, it is important to compare the phenotypes between parent plants and progeny plants, not the phenotypes between the grazed plants and CK. I did not see any information about the stomatal characteristics of parent plants in both grazed and fenced pastures. Therefore, it is not clear whether it is due to genetic performances of the plants or not. I suggested to add another experiment in the field to test the difference of stomatal characteristics of grazed plants and the CK. If it shows the same phenomena as the greenhouse experiments on bud. Then, you can say this is the legacy effects.
2. For the grazing experiment, I am not sure how you can define overgrazing? What is the criteria to rule the overgrazing in your study.
3. I am not clear what the linkages between the 8 genes measured and stomatal characteristics. You need to make clear frameworks/flowchart for the relationship between genes and the stomatal characteristics.
4. How can you draw the conclusion of down-regulated 2-18 times and up-regulated 3 times without any quantified analysis?
5. English is poor. There are many grammatical errors in the paper. Please find a native English speaker to make corrections.

Reviewer 2 ·

Basic reporting

I think the manuscript is clearly written and relevant in the context of stomata on the legacy effect of historical overgrazing on the grassland plant. abs the literature references were sufficient.and the structure were professional.

Experimental design

The experiment design and techniques employed are appropriate. Relevance and novelty of the study are obviously.

Validity of the findings

This paper is valuable on exploring the role of stomata on the legacy effect of historical overgrazing on the grassland plant. It will be helpful for us to better understand the possible mechanism how plant adapting to long term grazing.

Additional comments

Stoma is one of the most important organs for plant photosynthesis and water transportation. I think the manuscript is clearly written and relevant in the context of stomata on the legacy effect of historical overgrazing on the grassland plant. The experiment design and techniques employed are appropriate. Relevance and novelty of the study are obviously. This paper is valuable on exploring the role of stomata on the legacy effect of historical overgrazing on the grassland plant. It will be helpful for us to better understand the possible mechanism how plant adapting to long term grazing. The paper can be considered for publication after addressing the following questions.

Abstract,
1. The abstract is too long, there are so much words for the background. It is best to cut back and leave the important part.
“Grazing, one of the primary modes in the utilization of grassland, is the main cause of grasslands degradation. Long-term overgrazing results in dwarf phenotype of perennial plants and decreases photosynthesis, which is not only a contemporary phenomenon, but also sustainable in future generations. However, it is unknown what is the mechanism of legacy effect of lower photosynthesis from historical grazing. Stomata, as an important regulatory organ of photosynthesis, is one of key elements to control the efficiency of photosynthesis. What role stomata play in this low- photosynthesis induced by long-term overgrazing still have no answer”.
2. The Keywords not include the main key word. I suggest genes expression should be added in
Introduction
1. Line 39 (Lohmann et al., 2012; Wang, Wang & Wang, 2006; Sasaki et al., 2008)
Should be the order (Wang, Wang & Wang, 2006; Sasaki et al., 2008; Lohmann et al., 2012)
The reference are also mess and need to be revised.
Wang W Y, Wang Q J, Wang HC. 2006. The effect of land management on plant community composition, speciesdiversity, and productivity of alpine Kobersia steppe meadow. Ecological Research 21:181-187 DOI 10.1007/s11284-005-0108-z.
3. Line 80 Leymus chinensis, which is a perennial, rhizomatous clonal grass with high palatability and forage value (Liu, 2006), covers a large area of the eastern Eurasian temperate grassland.
The author should mention the importance of Leymus chinensis in the local grassland ecosystem.
4. The important and research of gene expression in grassland should be supplemented.
Materials and Methods
1. As to grazing treatments, 6 sheep were continuously grazing in the plot with the area of 0.64 hm2 during the growing season from May to October each year.
Grazing treatments should be converted into grazing intensity
2. As to plant materials, why the buds were taken and planted in greenhouse instead of taking plant materials in the field directly on summer?
3. The temperature was about 25℃ during the day and 15℃ at night. The phenotypic traits, gas exchange parameters, stomatal traits and related gene expression studies were tested successively after plant transplantation.
The exact test time should be stated.
4. As to real-time PCR analysis, the samples were taken from two treatments long-term enclosure and overgrazing? Are these two treatments the same with the Grazing and No grazing treatment? If not, please describe the difference?
5. As to real-time PCR analysis, how the 8 genes related to stoma development and regulation were selected?
Results
1. The authors mentioned the progeny developed from the grazed and un-grazed parents, How did the progeny obtained? Are they sexual or asexual progeny?
2. The authors should provide a microscopic stomatal structure picture, so it is very intuition for the readers
Discussion,
1. How to explain the up-regulation of cycp3L -1? Doe s the stoma data support this findings?
Conclusion
Line 274 which resulted in the dwarf phenotype of offspring plants (Fig. 6).
The (Fig.6) should be deleted

Annotated reviews are not available for download in order to protect the identity of reviewers who chose to remain anonymous.

Reviewer 3 ·

Basic reporting

The manuscript reports on the effects of grazing on a suite of leaf traits, with particular focus on stomatal density. The research is important and of interest. Overall the manuscript is coherent and well put together. However, it requires extensive editing for grammar and word usage.

Experimental design

The experimental design is appropriate. There is ambiguity in how plants were sampled in the field that requires more detail (e.g. how many samples, number per paddock, did they come from different parental plants, etc.).

Validity of the findings

The discussion needs revision. Because the research was done in the greenhouse and because there was little ancilliary data provided to explain some of the patterns much of the discussion is speculative. For example, water supply can have a significant impact on stomatal density and there is much speculation around this in the discussion, but not accompanying data - this is not necessarrily a problem, but the amount of discussion to this end needs to be reduced.

The mass of the leaves and the mass of the plants should have been measured. If this data is available it must be included with this manuscript. This would enable 1) better information on plant growth, as height is a poor indicator and 2) enable the authors to relate patterns to specific leaf area (SLA) a key metric for understanding plant growth in response to environmental change.

Wording throughout the manuscript around the idea of offspring and legacy effects needs to be written with clarity. This study uses clones from the parental plant and this must be made absolutely clear. The findings essentially represent phenotypic plasticity, an idea which is not dealt with to sufficient extent in this manuscript.

Additional comments

I have provided additional minor comments in the pdf.

Annotated reviews are not available for download in order to protect the identity of reviewers who chose to remain anonymous.

---

## Round 0.2 · accepted · Accept

Reviewers and I agree that the manuscript has adequately addressed all comments and suggestions made. I would like to congratulate them for the effort. This version of the manuscript is significantly improved.

Reviewer 2 ·

Basic reporting

I think the manuscript is clearly written and relevant in the context of stomata on the legacy effect of historical overgrazing on the grassland plant. abs the literature references were sufficient.and the structure were professional.

Experimental design

The experiment design and techniques employed are appropriate. Relevance and novelty of the study are obviously.

Validity of the findings

This paper is valuable on exploring the role of stomata on the legacy effect of historical overgrazing on the grassland plant. It will be helpful for us to better understand the possible mechanism how plant adapting to long term grazing.

Additional comments

This paper is valuable on exploring the role of stomata on the legacy effect of historical overgrazing on the grassland plant. It will be helpful for us to better understand the possible mechanism how plant adapting to long term grazing.

Reviewer 3 ·

Basic reporting

The authors have made appropriate changes to the manuscript and I have no concerns with their revisions.

The authors may find this paper relevant to their work:Oesterheld, M., McNaughton, S.J. Intraspecific variation in the response of Themeda triandra to defoliation: the effect of time of recovery and growth rates on compensatory growth. Oecologia 77, 181–186 (1988). https://doi.org/10.1007/BF00379184

Experimental design

No concerns

Validity of the findings

No concerns

Additional comments

Thank you for making revisions to the document.